# Complement Factor H and Related Proteins as Markers of Cardiovascular Risk in Pediatric Chronic Kidney Disease

**DOI:** 10.3390/biomedicines10061396

**Published:** 2022-06-13

**Authors:** Wei-Ting Liao, Wei-Ling Chen, You-Lin Tain, Chien-Ning Hsu

**Affiliations:** 1Division of Pediatric Nephrology, Kaohsiung Chang Gung Memorial Hospital, Kaohsiung 833, Taiwan; winona0409@cgmh.org.tw (W.-T.L.); weilingchen@cgmh.org.tw (W.-L.C.); tainyl@adm.cgmh.org.tw (Y.-L.T.); 2College of Medicine, Chang Gung University, Taoyuan 333, Taiwan; 3Department of Pharmacy, Kaohsiung Chang Gung Memorial Hospital, Kaohsiung 833, Taiwan; 4School of Pharmacy, Kaohsiung Medical University, Kaohsiung 807, Taiwan

**Keywords:** cardiovascular disease, chronic kidney disease, children, complement factor H, complement factor H-related protein, congenital anomalies of the kidney and urinary tract, hypertension

## Abstract

Cardiovascular disease (CVD) is the main cause of mortality among chronic kidney disease (CKD) patients, both in adults and in children. Hypertension is one of the risk factors of CVD. For early detection of subclinical CVD in pediatric CKD, 24 h ambulatory blood pressure monitoring (ABPM), cardiosonography, and arterial stiffness assessment were evaluated. CAKUT (congenital anomalies of the kidney and urinary tract) are the main etiologies of pediatric CKD. Previously, by a proteomic approach, we identified complement factor H (CFH) and related proteins differentially expressed between children with CAKUT and non-CAKUT CKD. In this study, we aimed to evaluate whether CFH, CFH-related protein-2 (CFHR2), and CFH-related protein-3 (CFHR3) were related to CVD risk in children with CKD. This study included 102 subjects aged 6 to 18 years old. The non-CAKUT group had higher plasma CFHR3 levels than the CAKUT group (*p* = 0.046). CFHR3 was negatively correlated with LV mass (*p* = 0.009). CFHR2 was higher in children with CKD with 24 h hypertension in the ABPM profile (*p* < 0.05). In addition, children with non-CAKUT CKD with day-time hypertension (*p* = 0.036) and increased BP load (*p* = 0.018) displayed a lower plasma CFHR3 level. Our results highlight that CFH and related proteins play a role for CVD in children with CKD. Early assessment of CFH, CFHR2, and CFHR3 may have clinical utility in discriminating CV risk in children with CKD with different etiologies.

## 1. Introduction

Considering that cardiovascular diseases (CVD) are the leading cause of death in children with CKD [1], assessing cardiovascular risks and detecting early markers of CVD are important. Unlike in adults, congenital anomalies of the kidney and urinary tract (CAKUT) are the most common etiologies of CKD in childhood [2]. Hypertension has long been a defined risk factor for both CVD and CKD. Over 50% of children with early-stage CKD had abnormalities on 24 h ambulatory blood pressure monitoring (ABPM) in our previous studies [3,4]. ABPM is a powerful tool for detecting missed hypertension and is strongly associated with known risk factors for end-stage kidney disease (ESKD) [5]. In addition, noninvasive assessments of arterial stiffness and left ventricular hypertrophy are also surrogate markers of early events of CVD. These include carotid intimal and medial thickness (cIMT), pulse wave velocity (PWV), ambulatory arterial stiffness index (AASI), left ventricular (LV) mass, and LV mass index (LVMI) [3,5,6]. As CAKUT often develops early in life and may progress to ESKD in childhood or early adulthood, early detection of the abovementioned CVD surrogate markers can help prevent further mortality and morbidity in pediatric CKD.

In addition to functional and structural surrogate markers, biomarkers are also important in early detection of CVD. Proteomics techniques have been used to identify various biomarkers of kidney diseases [7]. Combined with mass spectrometry, isobaric tags for relative and absolute protein quantification (iTRAQ) have been valuable in discovering potential protein biomarkers [8]. Our previous study using the iTRAQ technique identified several potential proteins expressed differentially between children with CKD with or without ABPM abnormalities, including complement factor H (CFH), CFH-related protein-2 (CFHR2), and CFH-related protein-3 (CFHR3) [9].

The complement system is a critical part of innate immunity to protect our body against infection and tumors [10]. Three main pathways, including the classical pathway, the lectin pathway, and the alternative pathway, contribute to complement activation. CFH is a main regulator of the alternative pathway to inactivate C3b and then to block the amplification loop of C3 convertases and the terminal pathway [11]. The CFHR proteins can be divided into two groups according to dimerization or not, as circulating in plasma: CFHR2 existing in dimeric form belongs to group 1, while group 2, including CFHR3, is oligomeric [12]. Considering the competition between CFHRs and CFH for binding to C3b, as described in [9], we assumed that simultaneous analysis of CFH, CFHR2, and CFHR3 might provide more information to reflect the homeostasis of the complement system.

In this study, we aimed to discover the association between CFH, CFHR2, CFHR3, and ABPM anomalies, parameters of cardiosonography, and arterial stiffness in children with CKD.

## 2. Materials and Methods

The protocol for this Precision Medicine Project of Pediatric Chronic Kidney Disease (PMP-PCKD) study was obtained with consent from the Institution Review Board and Ethics Committee of Chang Gung Medical Foundation, Taipei, Taiwan (201601181A3 and 201701735A3C501). Chronic kidney disease stage was defined by the KDIGO guidelines [13]. The Schwartz formula was used to calculate the estimated glomerular filtration rate (eGFR) [14]. The following were the inclusion criteria of the study: CKD patients aged 6–18 years. Exclusion criteria were patients who were pregnant, had CKD stage 5, had ESKD, were receiving a renal transplant, had congestive heart failure, or were unable to receive cardiovascular assessment. All of the participants, including their parents, provided signed consent to participate in the study. As the most common underlying disease in pediatric CKD is CAKUT [2], the etiologies of CKD were divided into two categories: CAKUT or non-CAKUT.

All participants received cardiovascular assessments, a fasting blood draw, and spot urine collection. Urine was stored in a −80 °C fridge until examination. The hospital central lab was responsible for measuring blood urea nitrogen (BUN), creatinine (Cr), uric acid, glucose, hemoglobin (Hb), hematocrit (A), sodium (Na), potassium (K), calcium (Ca), phosphate (P), urine total protein-to-Cr ratio (UPCR), triglyceride (TG), and low-density lipoprotein (LDL). Plasma CFH was examined using an ELISA kit (catalog number: CSB-E08931h, CUSABIO TECHNOLOGY LLC, Houston, TX, USA), CFHR2 was examined using an ELISA kit (catalog number: CSB-EL005275HU, CUSABIO TECHNOLOGY LLC, Houston, TX, USA), and CFHR3 was examined using an ELISA kit (catalog number: CSB-EL005276HU, CUSABIO TECHNOLOGY LLC, Houston, TX, USA). All three kits’ inter-assay coefficients of variations were <10%.

We measured office blood pressure and ABPM for all subjects in this study. Before measuring office BP, children rested for 5 min. Then, we recorded the seated non-dominant arm BP three times at one-minute intervals. ABPM was conducted on an Oscar II monitoring machine (SunTech Medical, Morrisville, NC, USA) for 24 h. The ABPM was equipped by a trained nurse as previously described [15]. The BP was measured every 20 min from 7 a.m. to 10 p.m. and every 30 min from 10 p.m. to 7 a.m. Moreover, the patients recorded the timing of waking, sleeping, and any activities that would fluctuate BP. Abnormal ABPM profiles were determined as follows: (1) daytime BP, nighttime BP, systolic BP (SBP), or diastolic BP (DBP) ≥ 95th percentile basis as per the ABPM reference [16]; (2) BP load ≥ 25% in daytime or nighttime SBP or DBP; (3) BP load decrease < 10% during sleep compared to the average awake BP load. In addition, the AASI (ambulatory arterial stiffness index) was counted by the formula of 1-DBP/SBP [13]. Cardiosonography was operated by a pediatric cardiologist with ultrasound (Philips, Bothell, WA, USA). The LV mass was calculated in the M-mode setting. The LVMI (left ventricular mass index) was also counted by dividing LV mass to height^3^ [17]. The PWV (pulse wave velocity), a marker of arterial stiffness, was examined by a ProSound7 ultrasound (ProSound7, e-tracking system; Aloka Co., Tokyo, Japan).

Categorical variables are described as numbers and percentages. Continuous variables are presented as medians and interquartile ranges (IQRs; 25th−75th percentile). We used the Mann–Whitney U-test or chi-square test to test the variables’ difference between two groups and Spearman’s rank correlation coefficient to identify the associations of variables. A linear and logistic regression model were also operated, followed by entering multivariable analyses for adjusting related parameters for CFH, CFHR2, CFHR3, and CV risk markers. A *p* < 0.05 was considered statistically significant. All analyses were conducted on Statistical Package for the Social Sciences (SPSS) software 22.0 (Chicago, IL, USA).

## 3. Results

### 3.1. Population Characteristics

A total of 102 subjects aged 6–18 years old were enrolled in the analysis. Table 1 shows the clinical, anthropometric, and biomedical characteristics of the participants. CAKUT accounted for 56.9% (58/102) of the etiologies. In the CAKUT group, renal agenesis accounted for 31% (18/58), renal dysgenesis was 16% (9/58), obstructive nephropathy was 10% (6/58), reflux nephropathy was 26% (15/58), multicystic dysplastic kidney disease was 10% (6/58), and prune-belly syndrome was 2% (1/58). The non-CAKUT group included 15 cases of glomerulonephritis, 13 cases of nephrotic syndrome, 7 cases of IgA nephropathy, 4 cases of lupus nephritis, 2 cases of purpura nephritis, 1 case of ANCA vasculitis, 1 case of PSGN, and 1 case of FSGS. There were 56 (55%) male participants. The median age was 10.8 years old in the CAKUT group and 12.8 in the non-CAKUT group. There were 62% and 80% CKD stage 1 in the CAKUT and non-CAKUT groups, respectively. In addition, the non-CAKUT group had a higher eGFR and UPCR and a lower Hct, fasting glucose, Ca, and P level. Neither systolic or diastolic BP nor office hypertension prevalence significantly differed between the CAKUT and non-CAKUT groups. Regarding immunosuppressant therapy, none of the CAKUT group received immunosuppressant therapy, while 47.7% children (21/44) in the non-CAKUT group received prednisolone alone or with cyclosporin.

### 3.2. Complement Factor H and Related Proteins

As shown in Table 2, there was no difference in plasma CFH and CFHR2 levels between the CAKUT and non-CAKUT groups. CFHR3 was higher in the non-CAKUT group compared to the CAKUT group. We found that CFH and related proteins were not correlated with eGFR (all *p* > 0.05) and UPCR (all *p* > 0.05). There was also no difference in plasma CFH and related proteins in non-CAKUT patients under different immunosuppressant therapy (Appendix A).

### 3.3. Ambulatory Blood Pressure Monitoring and Cardiovascular Assessment

Compared to the office BP, the abnormal ABPM profile was more prevalent (76% vs. 30%). Patients had at least one abnormality in the ABPM, including 19 subjects (19%) with 24 h hypertension, 11 subjects (11%) with daytime hypertension, 24 subjects (24%) with nighttime hypertension, 58 subjects (57%) with increased BP load, and 58 subjects (57%) with non-dipping nocturnal BP. However, the ABPM profile did not differ between the CAKUT and non-CAKUT groups. Additionally, there was no difference in cardiovascular assessments between the CAKUT and non-CAKUT groups, including the LV mass, LVMI, PWV, cIMT, and AASI.

### 3.4. Association between Cardiovascular Assessment and Complement Factor H and Related Proteins

Table 3 illustrates the coefficient of correlation between CV assessments and CFH and related proteins in children with CKD. In the CAKUT group, CFH had a positive correlation with the LV mass and LVMI. CFHR3 was negatively correlated with the LV mass, LVMI, and PWV in the non-CAKUT group but positively correlated with the LVMI in the CAKUT group.

### 3.5. Association between Ambulatory Blood Pressure Monitoring and Complement Factor H and Related Proteins

We also analyzed the plasma levels of CFH and related proteins in children with CKD and determined their associations with the ABPM profile (Appendix A). We observed a higher CFHR2 level in children with CKD with an abnormal 24 h BP compared to a normal ABPM profile (118.4 vs. 96.8 μg/mL, *p* < 0.05). In the CAKUT group, the plasma levels of CFH and related proteins were not different between children with abnormal and normal ABPM profiles (Appendix A).

However, Figure 1 shows that the children with non-CAKUT CKD with a high daytime BP or a high BP load had lower CFHR3 levels compared to those with a normal ABPM profile (Appendix A).

Associations between CV risk and CFH and related proteins were further examined in a multivariate linear regression or logistic model (Table 4, Table 5 and Table 6). We performed multivariate linear regression model by entering confounding factors of age, sex, BMI, eGFR, TG, and LDL. The LV mass was positively associated with CFHR2 in the children with CKD group (Table 4). In the CAKUT group, CFH was positively correlated to the LVMI, controlling for other factors (Table 5). In the non-CAKUT group, the regression model revealed significantly positive associations between CFHR2, LV mass, and LVMI, whereas CFHR3 had negative associations with the LV mass and LVMI (Table 6).

As for the multivariate logistic regression model, after adjusting for age, BMI, eGFR, TG, and LDL, CFHR2 was associated with 24 h hypertension and nighttime hypertension in the CKD group (Table 4). Additionally, CFHR2 was associated with increased BP load in the CAKUT group (Table 5) but not in the non-CAKUT group (Table 6).

## 4. Discussion

The major findings of this study were as follows: (1) the use of office BP alone would have missed the detection of hypertension in more than half the children; (2) children with CKD with non-CAKUT etiology had a higher plasma CFHR3 level compared to the CAKUT group; (3) CFHR3 was negatively correlated with the LV mass and LVMI in the non-CAKUT group, whereas CFH was positively correlated with the LV mass and LVMI in the CAKUT group; (4) CFHR2 was associated with 24 h hypertension and nighttime hypertension detected by ABPM in children with CKD; and (5) a lower plasma CFHR3 level was noted in the non-CAKUT group with daytime hypertension and an increased BP load. We present a brief summary of their relationships in Figure 2.

In the current study, the non-CAKUT group had a higher eGFR and UPCR. As the normal eGFR number is defined as more than 90 mL/min/1.73 m^2^ [13], both groups actually displayed normal renal function. Most non-CAKUT children had glomerular diseases associated with proteinuria, which is caused by increased glomerular permeability. Our results in the non-CAKUT group were consistent with previous studies showing a higher UPCR than children with CAKUT [2,4].

We also noted that there is lack of correlation between CFH proteins and eGFR and UPCR. Although a previous study revealed that urinary CFH protein excretion correlated negatively with eGFR [18], whether urinary excretion of CFH protein increases with declining GFR due to increased glomerular permeability or impaired tubular reabsorption remains unknown. In addition, proteinuria is a well-known marker for progression of CKD, independent of the underlying kidney disease. Though increasing evidence supports the role of complement in proteinuria-mediated kidney damage [19], plasma CFH protein level is not correlated with UPCR or eGFR in this study, most likely because most children with CKD are still in the early stages.

Prior research has demonstrated that ABPM is superior to office BP measurement in its ability to distinguish patients at high risk for target-organ damage and has a strong relationship to LVM [20]. Although our results revealed a high prevalence of BP abnormalities of ABPM, there was no difference regarding an abnormal ABPM profile between the CAKUT and non-CAKUT group. Currently no studies report the association between CFH and related proteins with cardiovascular risk in pediatric CKD. To the best of our knowledge, we are the first to report the impact of CFH and related proteins on the CVD risk in children with CKD.

CFH plays a key role as a regulator in the alternative pathway of the complement system. CFH participates in inflammation and is associated with the pathogenesis of many diseases, including CVD [21,22]. Previous studies have shown an association between CFH genetic polymorphism and hypertension [23], indicating that CFH may be a significant predictor of CVD [24]. In support of the notion that CFH is related to CVD risk, our results demonstrated that CFH had a positive correlation with the LV mass and LVMI in children with CKD.

The precise role of CFHR proteins is still unclear as to the inhibitory effect thought initially, which is in contrast to recent studies reporting the enhancing activation of the complement pathway [11]. In our data, CFHR2 had a positive correlation with the LV mass, 24 h hypertension, and nighttime hypertension in children with CKD. In the CAKUT group, CFHR2 was also associated with an increased BP load. CFHR2 is considered an inhibitory regulator of the C3 alternative pathway convertase, acting simultaneously with CFH with a synergic suppression effect [25]. Another study revealed that a higher urine CFHR2 level was associated with a greater risk of death in patients with proteinuric diabetic kidney disease [26]. As shown in Figure 1, we, therefore, hypothesized that CFHR2 may play a role in the pathogenesis of CVD in pediatric CKD and may serve as a predictor of early CVD risk, especially in the CAKUT group.

Regarding CFHR3, it presents homologous domains of CFH, sharing some binding characteristics of CFH and, thus, was considered to have CFH cofactor activity [11]. The circulating CFHR3 level was greatly influenced by the gene deletion of the CFHR3-CFHR1 allele and gene variants of the CFHR3 gene [10]. Polymorphism of the CFHR3 gene is associated with an increased risk of atypical hemolytic uremic syndrome [10,27]. In line with previous studies showing that a higher CFHR3 level in lupus nephritis was positively correlated with SLEDAI values [28], our results indicated that the CFHR3 level was higher in the non-CAKUT group.

Interestingly, our study revealed a negative correlation of CFHR3 with the LV mass, LVMI, daytime hypertension, and increased BP load in children in the non-CAKUT group. Whether high CFHR3 reduces CV risks or CVD induces compensatory increases in CFHR3 deserves further elucidation. Additionally, future studies are required to elucidate how CFHR3 works differentially in CAKUT and non-CAKUT CKD and to determine their underlying mechanisms. Our data support the notion that early evaluation of CFH, CHHR2, and CFHR3 in children with CKD will have clinical utility in discriminating CV risk. Although the proportion of abnormal BP load and CV risk markers did not differ between the CAKUT and non-CAKUT group, CFH-related markers may differentially predict CV risks in these two groups that have somewhat different etiologies.

Our study had some limitations. Firstly, we only studied CFH, CFHR2, and CFHR3 levels at one point of time, although there might be serial changes of these proteins at different time points, and there are other proteins in the CFH-related protein family. As immunosuppressive therapy could affect these proteins [29], whether serial analyses of CFH-related proteins are helpful in monitoring disease activity or therapeutic response deserves further elucidation. Considering different CFHRs can compete with CFH for binding, further research is warranted to investigate the individual role of the CFH-related protein family in children with CKD. Secondly, we did not determine the levels of CFH, CFHR2, and CFHR3 in the normal child population. Lack of normal reference and the pathologic cut point value of the proteins may limit further clinical translation. Thirdly, our data were from a single-centered cohort study; whether children with CKD from one hospital can represent the whole pediatric CKD population remains uncertain. Further multicenter studies are needed to support our findings. Last, the group of diseases encompassed by non-CAKUT did not have a single disease mechanism. Although complement-mediated kidney diseases belong to the non-CAKUT category [30], our results indicate that plasma CFH-related protein levels vary in different etiologies of the non-CAKUT group. Thus, further subgroups in larger cohorts by different etiologies in the non-CAKUT category will improve the prevention and treatment of CKD, with the aid of CFH-related proteins.

## 5. Conclusions

Our results highlight that CFH and related proteins play a role for CVD in children with CKD. Early assessment of a CVD surrogate marker, especially ABPM, may increase early detection of CVD risk in pediatric CKD. Importantly, early measurement of CFH, CFHR2, and CFHR3 may have clinical utility in discriminating CV risk in children with CKD with different etiologies.

## Figures and Tables

**Figure 1 biomedicines-10-01396-f001:**
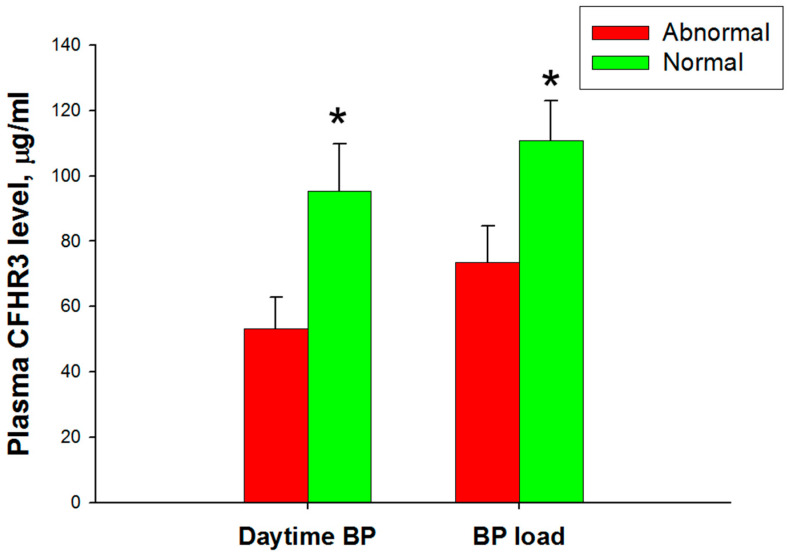
Plasma CFHR3 level in children with non-CAKUT CKD when subgrouped to an abnormal ABPM profile in daytime BP (left) and BP load (right). * *p* < 0.05 vs. abnormal ABPM by the Mann–Whitney U-test.

**Figure 2 biomedicines-10-01396-f002:**
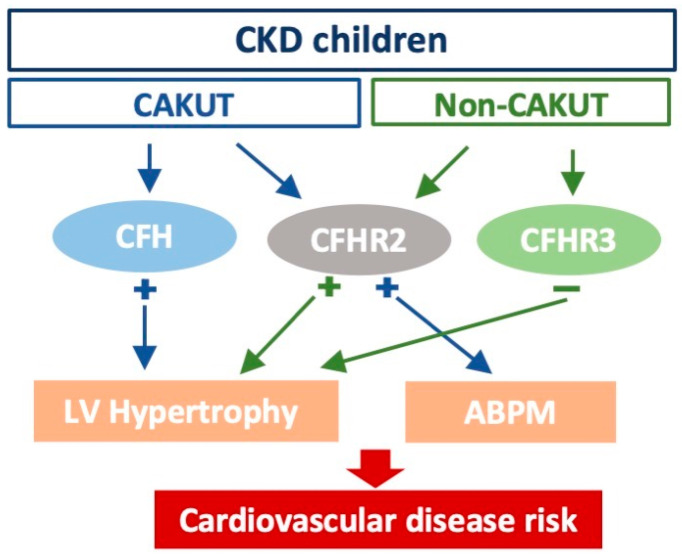
CFH and CFHR2 had a positive correlation with the LV mass, LVMI, and an abnormal ABPM profile in children with CKD and when subgrouped to the CAKUT population; CFHR3 had a negative correlation with the LV mass, LVMI in the non-CAKUT group.

**Table 1 biomedicines-10-01396-t001:** Clinical, anthropometric, and biomedical characteristics in children with CKD.

Group	CAKUT	Non-CAKUT
	*n* = 58	*n* = 44
Sex M:F	35:23	21:23
CKD stage 1	36	35
CKD stage 2,3,4	22	9
Age	10.8 (9.0–14.2)	12.8 (8.6–15.6)
Body weight, percentile	50 (22.5–85)	50 (15–85)
Body height, percentile	50 (25–85)	50 (25–75)
Systolic blood pressure, mmHg	114 (105–122.5)	109.5 (101–122)
Diastolic blood pressure, mmHg	71 (64–76)	68 (64–79.5)
Body mass index, kg/m^2^	18.1 (15.8–22.1)	19.2 (16.8–22.6)
Hypertension (office BP)	17 (29.3%)	14 (31.8%)
Immunosuppressant therapy		
None	58 (100%)	23 (52.3%) *
Prednisolone	0	17 (38.6%)
Prednisolone + cyclosporin	0	4 (9.1%)
Blood urea nitrogen, mg/dL	13 (11–16)	12 (10–15.75)
Creatinine, mg/dL	0.6 (0.49–0.8)	0.52 (0.46–0.72)
eGFR, mL/min/1.73 m^2^	99.4 (82.4–114.3)	117.1 (91.5–130.1) *
Urine total protein-to-creatinine ratio, mg/g	49.0 (34.6–71.5)	225.4 (56.6–938.8) *
Hemoglobin, g/dL	13.8 (13–14.6)	13.4 (12.4–14.2)
Hematocrit, %	40.6 (38.9–43.0)	39.2 (37.3–41.4) *
Fasting glucose	89 (86–93)	87 (82–92) *
Uric acid, mg/dL	5.3 (4.3–6.3)	5.6 (4.2–7.1)
Sodium, meq/L	141 (140–142)	141 (139–142)
Potassium, meq/L	4.3 (4.1–4.5)	4.3 (4.1–4.5)
Calcium, mg/dL	9.8 (9.5–10.1)	9.5 (9.1–9.8) *
Phosphorus, mg/dL	4.9 (4.6–5.2)	4.7 (4.2–4.9) *

Data are medians (25th, 75th percentile) or *n* (%). * *p* < 0.05 for CAKUT vs. non-CAKUT by the chi-square test or Mann–Whitney U-test. BP = blood pressure; CAKUT = congenital anomalies of the kidney and urinary tract; eGFR = estimated glomerular filtration rate.

**Table 2 biomedicines-10-01396-t002:** Plasma complement factor H and related protein level in children with CKD.

Group	CAKUT	Non-CAKUT
	*n* = 58	*n* = 44
CFH (μg/mL)	706.7 (478.0–1021.0)	635.9 (317.0–1013.9)
CFHR2 (μg/mL)	105.1 (73.0–144.1)	108.9 (75.6–126.4)
CFHR3 (μg/mL)	60.4 (37.3–92.4)	77.5 (50.2–127.7) *

Data are medians (25th, 75th percentile) or *n* (%). * *p* < 0.05 by the Mann–Whitney U-test.

**Table 3 biomedicines-10-01396-t003:** Coefficient of correlation of CFH, CFHR2, CFHR3 vs. CV assessments in the CAKUT and non-CAKUT groups.

Group	CAKUT			Non-CAKUT		
	CFH	CFHR2	CFHR3	CFH	CFHR2	CFHR3
LV mass	0.422 **	−0.065	0.125	0.141	0.266	−0.462 **
LVMI	0.322 *	−0.052	0.302 *	0.087	0.221	−0.392 **
PWV-beta	−0.203	0.164	0.066	−0.235	−0.151	−0.446 **
cIMT	−0.097	0.099	−0.028	−0.068	−0.113	−0.127

* *p* < 0.05; ** *p* < 0.01 by Spearman’s rank correlation.

**Table 4 biomedicines-10-01396-t004:** Regression model of cardiovascular risks and CFH-related proteins in children with CKD.

Children with CKD (*n* = 102)					
Dependent Variable	Explanatory Variable	Adjusted^a^		Model	
		Beta	*p*-value	R^2^	*p*-value
LV mass	CFHR2	0.12	0.036 *	0.718	<0.001
		OR	*p*-value		
24 h hypertension	CFHR2	1.019	0.032 *		
Nighttime hypertension	CFHR2	1.019	0.02 *		

OR: odds ratio; adjusted^a^ for age, sex, BMI, eGFR, TG, and LDL; * *p* < 0.05.

**Table 5 biomedicines-10-01396-t005:** Regression model of cardiovascular risks and CFH-related proteins in the CAKUT group.

CAKUT (*n* = 58)					
Dependent Variable	Explanatory Variable	Adjusted^a^		Model	
		Beta	*p*-value	R^2^	*p*-value
LVMI	CFH	0.369	0.009 **	0.275	0.001
		OR	*p*-value		
Increased BP load	CFHR2	1.020	0.027 *		

OR: odds ratio; adjusted^a^ for age, sex, BMI, eGFR, TG, and LDL; * *p* < 0.05, ** *p* < 0.01.

**Table 6 biomedicines-10-01396-t006:** Regression model of cardiovascular risks and CFH-related proteins in the non-CAKUT group.

Non-CAKUT (*n* = 44)					
Dependent Variable	Explanatory Variable	Adjusted ^a^		Model	
		Beta	*p*-value	R^2^	*p*-value
LV mass	CFHR2	0.230	0.005 **	0.774	<0.001
	CFHR3	−0.233	0.015 *	0.76	<0.001
LVMI	CFHR2	0.320	0.007 *	0.518	<0.001
	CFHR3	−0.302	0.031 *	0.481	<0.001

Adjusted^a^ for age, sex, BMI, eGFR, TG, and LDL; * *p* < 0.05, ** *p* <0.01.

## Data Availability

Data are contained within the article.

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
