# Peer review of "Complement Factor H and Related Proteins as Markers of Cardiovascular Risk in Pediatric Chronic Kidney Disease"

_biomedicines, 2022, doi:10.3390/biomedicines10061396_

Round 1

Reviewer 1 Report

In the present study, the authors evaluated the association between cardiovascular risk markers (namely hypertension and LVMI) and regulatory proteins of the alternate complement pathway (namely CFH, CFHR2 and CFHR3) in pediatric patients with chronic kidney disease, in order to establish a causal relationship between the levels of complement proteins evaluated and the cardiovascular risk of the population studied.

Major issues

- It is unclear why the authors divided pediatric patients with CKD into two groups, CAKUT and non-CAKUT. The authors do not report on the existence of differences in cardiovascular risk markers between the two groups that could justify their analysis independently. Although the authors did not report on LVMI, nor on the LV mass of the two groups, they mentioned that systolic BP, diastolic BP and the prevalence of office hypertension were not different between the two groups;

- It is simply not possible to conclude the suggested causal relationship between the levels of complement proteins evaluated and the cardiovascular risk of the two populations, through the observed associations. How do the authors explain the causal relationship they suggest of the positive association between CFH levels ("a main regulator of alternative pathway to inactivate C3b and then to block the amplification loop of C3 convertases and the terminal pathway") and LVMI in the CAKUT group? If the negative relationship of CFHR3 with LV mass and LVMI in non-CAKUT group had a causal significance, then it would be expected that the CAKUT group, which had lower CFHR3 levels than the non-CAKUT group, would have greater LVMI and LV mass than the CAKUT group. However, this is not mentioned.

Minor

- It is difficult to explain that the eGFR of both groups has been greater than 90 ml/min, when 22/58 (~40%) of patients in the CAKUT group had CKD stages 2, 3 and 4, that is, less than 90 ml/min.

Reviewer 2 Report

The manuscript by Liao et. al. measures ambulatory blood pressures in a CKD pediatric cohort, to link abnormalities with in BP and cardiovascular disease risk with different levels of complement factor H proteins. Their key finding is that CFH proteins are associated with CVD risk in children with CKD, suggesting as potential use of such proteins a biomarkers in different etiologies. The manuscript is well written and the results are exiting, however, the presentation of the data is tedious for two main reasons, 1st) extensive use of acronyms and 2nd) all data is presented in tables. Some suggestions follow to improve the manuscript.

Comments:

The use of ‘sex’ is preferred over ‘gender’.

Refrain from calling glucose “sugar”.

Data presentation should be in a more visual way, which could either replace or complement tables. Figures to be created should be self-standing and contain a legend explaining its contents and defining all acronyms.

For future studies, I suggest the use of an activity tracker in conjunction with ABPM to have more objective data that could explain changes in blood pressure, as oppose to rely on patient self-reporting. Such monitors could also track pO2 levels that could also be informative to explain alteration in the nocturnal deep in BP.

Round 2

Reviewer 1 Report

The authors reviewed the manuscript and responded appropriately to the comments made.

Author Response

Thank you very much for your valuable comments.

Reviewer 2 Report

It this revised version of the manuscript, the Authors have addressed two minor comments from previous round. The data presentation on tables is still something to work on, in particular tables 4 and 5, which are very heavy to read and only show that CFHR3 is lower in non-CAKUTs with higher day-time BP or BP load. Maybe tables should go as supplementary data and these finding be highlighted in main manuscript as graphs.

Other comments:

Authors should present a rationale for grouping different etiologies in the nonCAKUT cohort. And discuss the possible implications on different pathobiological mechanisms in the findings of this study.

Table 1 should also contain whether patients were subject to immunosuppressive therapy at the time of the study. Discuss whether such therapy could affect the expression of the proteins studied.

I would be interesting to see the proteomic results parsed by actual etiology (maybe on supplemental table). Numbers may be low to do statistics but a visual of the data could still be informative.

The lack of correlation between CFH proteins and eGFR and UPCR should be discussed, and these two parameters are different between groups. 

Author Response

Thank you for valuable comments.

Round 3

Reviewer 2 Report

Great work addressing comments.

thanks